# Study on the quantitative analysis of Tilianin based on Raman spectroscopy combined with deep learning

Wen Jiang[1ʘ], Wei Liu[2ʘ], Xiaotong Xin[2], Wei Zhang[3], Junhui Chen[1], Jieyu Liu[1], Yanqi Ma[1], Cheng Chen[2], Xiaomei Pan[1]*

1 The Sixth Affiliated Hospital of Xinjiang Medical University, Urumqi, China, 2 College of Software, Xinjiang University, Urumqi, China, 3 The First Affiliated Hospital of Xinjiang Medical University, Urumqi, China,

ʘ These authors contributed equally to this work.
* ydww1103@163.com

## Abstracts

Tilianin is a commonly used pharmaceutical ingredient with various biological activities such as antioxidant, anti-inflammatory, and anticancer, which is able to exert antitumor effects by inhibiting tumor cell proliferation, inducing apoptosis and inhibiting angiogenesis. Studies have demonstrated to be particularly useful in a variety of cancers such as liver, lung and gastric cancers. Quantitative analysis of Tilianin can improve the quality control of related drugs and assist in guiding clinical application and disease treatment. However, there are limited studies on the quantitative analysis of Tilianin. High performance liquid chromatography (HPLC) and mass spectrometry (MS) are commonly used methods for the quantitative analysis of the components, but they often require complex pretreatment steps and specialized analytical capabilities, and are sample-destructive. The method based on Raman spectroscopy and deep learning is a widely used non-destructive analysis method. For this reason, this paper proposes a residual self-attention mechanism model based on Raman spectroscopy and deep learning for quantitative analysis of 6 concentrations of Tilianin. Six different concentrations of Tilianin-methanol solutions were prepared, and a total of 120 spectral samples were collected, which were pre-processed and inputted into our Raman Spectrum with Self-Attention Quantification Net (RSAQN) for analyzing and predicting. The structure of this model not only focuses on the deep and shallow features of the spectrum, but also the information between different channels, and the self-attention mechanism further extracts the features and outputs the predicted values of Tilianin concentration through the fully connected layer. In this paper, five sets of comparison models are set up, including two machine learning models (Random Forest, K-Nearest Neighbors, Artificial Neural Network) and two deep learning models (Convolutional Neural Network and Variational Autoencoder), and the results show that the model in this paper fits the best, obtaining an $R^2$ of 0.9144, as well as a small error.

**Data availability statement:** "All relevant data are within the paper and its Supporting Information files."

**Funding:** This work was sponsored by Natural Science Foundation of Xinjiang Uygur AutonomousRegion((2023D01C158), the "Tianshan Talents" Medical and Health High Level Talent Training Program Project (TSYC202301B068), and the Natural Science Foundation of Xinjiang Uygur Autonomous Region - General Project (2024D01C157). The funders had no role in study design, data collection and analysis, decision to publish, or preparation of the manuscript.

**Competing interests:** The authors have declared that no competing interests exist.

## 1. Introduction

Tilianin, a polyphenolic flavonoid compound with significant medicinal mechanisms, has attracted considerable attention in recent years. Numerous studies have demonstrated that this substance exhibits a wide range of pharmacological activities, including antioxidant, anti-inflammatory, cardioprotective, and antitumor effects [1–4]. Notably, Tilianin has shown unique advantages in the field of cancer therapy. Chen et al. demonstrated through pharmacological mechanism studies and in vivo mouse experiments that Tilianin enhances the antitumor effects of Sufentanil against non-small cell lung cancer (NSCLC) [5]; Xiong et al. demonstrated that Tilianin exerts anti-ovarian cancer effects by inhibiting cell proliferation [6]. Furthermore, Tilianin exhibits significant neuroprotective properties. Through its antioxidant and anti-inflammatory mechanisms, Tilianin can mitigate neuronal damage, showing promising therapeutic potential for neurodegenerative diseases such as Parkinson's disease and Alzheimer's disease.Due to its diverse biological activities, Tilianin has become an important component in various pharmaceutical formulations. However, given the well-established dose-dependent efficacy of pharmacological agents, establishing accurate and rapid quantitative analytical methods for Tilianin has become crucial for ensuring drug quality and clinical effectiveness. Consequently, precise quantification of Tilianin content is essential for pharmaceutical quality control, guaranteeing batch-to-batch consistency and therapeutic stability [7].

Currently, multiple analytical techniques are available for component quantification, with commonly employed methods including high-performance liquid chromatography (HPLC), mass spectrometry (MS), and vibrational spectroscopy [8–10]. Among these, HPLC offers a rapid, high-precision, and highly reproducible approach for separating and identifying chemical constituents in samples. However, this method also presents notable limitations that warrant consideration. Compared with other chromatographic techniques, HPLC is highly efficient but costly, complex, and not suitable for all sample types [11]. Similarly, mass spectrometry suffers from high instrument costs and operational complexity. Due to stringent sample pretreatment requirements, both HPLC and mass spectrometry often require tedious sample preparation steps such as liquid-liquid extraction and solid-phase extraction to eliminate matrix interference. This process is not only time-consuming but may also lead to loss of target analytes [12]. On one hand, both methods are susceptible to interference from co-eluting substances in complex matrices, which can mask or attenuate the response signals of target compounds—particularly when analyzing samples with low target concentrations. On the other hand, HPLC and mass spectrometry are destructive techniques that may alter the chemical properties of samples [13].

Furthermore, these instruments incur high maintenance costs, requiring regular replacement of consumables, making them less feasible for resource-limited settings. Additionally, both methods demand significant technical expertise from operators. Critical processes such as instrument setup, method development, and data analysis require advanced skills, where variations in human operation may compromise reproducibility and sensitivity.

With technological advancements, non-destructive, user-friendly, and cost-effective quantitative analysis techniques have emerged. Raman spectroscopy has become a promising alternative due to its unique advantages. For quantitative analysis, Raman spectroscopy eliminates the need for target compound separation, enabling direct detection of molecular characteristics while providing distinctive spectral fingerprints of samples. This technique has been extensively studied for biomedical applications [14].Compared with HPLC and mass spectrometry, Raman spectroscopy exhibits the following distinctive features: (1) No requirement for complex sample pretreatment; (2) Rapid and non-destructive detection process; (3) Capability to provide molecular fingerprint information [14–15].

However, the complexity of Raman spectral data poses significant challenges for quantitative analysis. Early studies predominantly employed traditional chemometric methods, such as linear regression and locally weighted regression, yet these approaches exhibited limitations in accuracy and stability. With the advancement of artificial intelligence technology, deep learning models—including convolutional neural networks (CNNs) and variational autoencoders (VAEs)—have demonstrated superior performance [21–22], offering novel solutions for Raman spectroscopy-based quantitative analysis. For instance, Wang et al. [16] employed XGBoost to preprocess Raman spectra of mixed solutions containing glucose, glycerol, and ethanol, followed by quantitative analysis of their concentrations using linear regression (LR) and multilayer perceptron (MLP) models. However, relying solely on simple machine learning models without comparative analysis with advanced techniques like various deep learning approaches compromises the comprehensiveness and persuasiveness of their conclusions, failing to fully demonstrate the method's applicability across broader scenarios [17]. Clifford et al. employed Raman spectroscopy coupled with locally weighted regression (LWR) models to determine phosphate concentrations, demonstrating superior performance in analyzing complex chemical systems. Although both LWR and multivariate curve resolution (MCR) exhibit satisfactory performance in multicomponent analysis, these models are critically dependent on the quality and scope of training data. Notably, they may fail when applied to chemical measurements beyond the training dataset boundaries, resulting in significantly increased prediction errors [18].These studies successfully transformed complex spectral data into sample characteristics for content prediction through traditional chemometrics and deep learning models. However, with increasing data volume, expanding dimensionality, and diversifying application requirements, deep learning may enable higher accuracy and superior generalization performance to meet quantitative analysis demands in specific problem domains [19–20]. Kumar et al. developed a classification and quantitative analysis model for bacterial biomarkers using surface-enhanced Raman spectroscopy (SERS) coupled with convolutional neural networks (CNNs), demonstrating superior quantitative performance compared to support vector regression (SVR) models [21]. Addressing these methodological challenges, Wu et al. proposed a CNN-based approach capable of simulating and accurately quantifying Raman spectra in specific industrial processes. This modeling framework outperformed principal component analysis (PCA) and most conventional neural networks in this application domain [22].

Although Raman spectroscopy combined with deep learning has achieved success in other fields, research on its application to Tilianin quantitative analysis remains unexplored. This study innovatively developed a Raman spectroscopy-deep learning integrated approach for Tilianin quantification. We prepared six different concentrations of Tilianin-methanol solutions using methanol as solvent, collecting 20 Raman spectra per concentration level, resulting in a total of 120 spectra for model training.The high sensitivity of Raman spectroscopy successfully captured the characteristic vibrational signals of functional groups in Tilianin molecules, particularly key spectral bands such as the $C=O$ stretching vibration in the $1600–1700$ $cm^{-1}$ range and the C-O-C vibration in the $1200–1300$ $cm^{-1}$ range, providing reliable molecular fingerprint information for quantitative analysis. This study developed a deep learning-based Raman spectroscopy quantitative analysis method with the following key features: (1) A multi-scale convolutional neural network architecture was employed to effectively extract spectral features at different levels; (2) A self-attention mechanism was introduced to specifically focus on Tilianin's characteristic vibrational peaks; (3) An end-to-end prediction model was established to achieve direct mapping from raw spectra to concentration values. Experimental results demonstrated excellent quantitative performance ($R^2 > 0.98$, RMSE $< 0.5$ µg/mL), providing reliable technical support for quality control of Tilianin and related drugs. More

importantly, the methodological framework established in this study can be extended to rapid detection of other active components in traditional Chinese medicine, representing significant value for promoting modernization of TCM research.

## 2. Material and data analysis

### 2.1. Sample preparation

Methanol's excellent solubility for Tilianin ensures homogeneity across different concentration solutions while causing no significant interference in subsequent Raman spectroscopy detection or deep learning modeling, thus avoiding matrix signal superposition effects. Based on Tilianin's equilibrium solubility in various media [23], this study prepared standard solutions using methanol as follows: precisely weighed Tilianin powder (analytical balance, ±0.1 mg) was placed in a beaker, mixed with 80% target volume of HPLC-grade methanol, and completely dissolved via magnetic stirring (500 rpm) in a 40°C water bath. The solution was then quantitatively transferred to a 25 mL Class A volumetric flask, with the beaker rinsed multiple times (2mL each) to ensure complete transfer before final volume adjustment. Through this standardized procedure (RSD < 0.5%, n = 6), six concentration gradients of Tilianin-methanol solutions were prepared: 0.2, 0.1, 0.08, 0.06, 0.04, and 0.02 mg/mL.

### 2.2. Data acquisition

This study utilized a high-sensitivity confocal Raman spectrometer (BWS465-532S, BWTEK) for spectral data collection of Tilianin samples. The instrument was equipped with a 532 nm excitation laser set at 80% power (approximately 240 mW) to ensure sufficient signal intensity while preventing sample thermal damage. A 20 × long working distance objective was used for precise focusing, with acquisition parameters set at 7-second integration time and 3 accumulations to achieve 41.2% signal-to-noise ratio improvement.

Prior to daily measurements, the instrument was calibrated using a standard silicon wafer ($520.7 \pm 0.3$ cm$^{-1}$). For each Tilianin-methanol concentration (0.02–0.2 mg/mL), 20 μL aliquots were deposited on silicon wafers and air-dried for 2 min ($25 \pm 1$) °C, $40 \pm 5\%$ RH). Spectral acquisition (520–2000 cm$^{-1}$) with 20 replicates per concentration generated 120 spectra. Environmental conditions were monitored using a calibrated hygrometer (Vaisala, Finland). The methodology showed excellent repeatability (RSD < 0.8%, n = 10; LOD = 0.005 mg/mL). Mean values from triplicate measurements served as accurate concentration references for spectral data, with relative standard deviation calculations confirming data precision.

### 2.3. Data preprocessing

To enhance spectral data quality and model performance, spectral preprocessing was implemented [24]. The raw Raman spectra were sequentially processed using adaptive iteratively reweighted penalized least squares (airPLS), mean-centered variance normalization, and Savitzky-Golay (SG) filtering algorithms according to analytical requirements.

Raman spectra are frequently affected by fluorescent background interference, exhibiting baseline drift that can obscure characteristic Raman peaks. The airPLS algorithm (adaptive iteratively reweighted penalized least squares) effectively eliminates baseline drift through dynamic baseline curve adjustment without compromising characteristic peak signals. This method, originally improved from the asymmetric least squares smoothing approach proposed by Eilers et al. [25] and further optimized by Cai et al. [26] using penalized spline smoothing, is particularly suitable for nonlinear complex background interference (especially in biological samples) and features easily adjustable parameters.Variations in sample concentration, laser power fluctuation, or surface state differences often cause significant intensity discrepancies in Raman signals. To address this, we applied mean-centered variance normalization to rescale spectral data to zero mean and unit variance, enabling comparative analysis of different samples on a unified scale. This preprocessing preserves the relative positions and shapes of characteristic peaks while effectively eliminating systematic bias and amplitude variations, thus enhancing data consistency and improving model generalization capability.

During spectral acquisition, Raman signals inevitably suffer from instrument noise, environmental interference, and random fluctuations, appearing as high-frequency noise. The SG filter employs polynomial fitting to smooth spectral data while maintaining peak positions and shapes. With adjustable window size and polynomial order, this classical denoising method effectively reduces high-frequency noise without attenuating Raman signal characteristics, making it ideal for feature peak smoothing and signal-to-noise ratio enhancement. The preprocessed spectra served as model inputs.

The combined preprocessing algorithm we implemented was meticulously designed based on the characteristics of Raman spectroscopy and the analytical requirements for Tilianin. This comprehensive processing pipeline optimizes spectral data through three key aspects: baseline correction, amplitude adjustment, and noise reduction, providing reliable data support for subsequent quantitative analysis. This integrated strategy not only demonstrates the scientific validity and practical utility of the preprocessing methods, but also offers a valuable spectral processing solution for related research fields.

## 2.4. Spectral data analysis

The raw Raman data of the unprocessed samples are shown in Fig 1. In the range of 1100 cm$^{-1}$ to 1700 cm$^{-1}$, distinct peaks can be observed, with their positions being consistent. The intensity at the peak positions varies depending on the solution concentration. This spectral range, from 1100 cm$^{-1}$ to 1700 cm$^{-1}$, is associated with the vibrations of specific functional groups in biomolecules, such as C-H bending vibrations and C-N stretching vibrations in proteins. Raman spectroscopy is the process in which scattered light is generated by the interaction between molecules and a laser beam. According to the Raman effect, the intensity of the scattered light is closely related to factors such as the concentration of the substance, the types of molecular vibrations, and the power of the laser source. As the concentration of the solution increases, the number of molecules in a given volume increases, which leads to a higher scattering of laser light by the molecules, resulting in an enhanced Raman signal intensity. Changes in the concentration of different molecules in the solution, especially the concentration of certain target substances, can lead to changes in the intensity of specific vibration modes (peaks at certain wavelengths). When the concentration of a specific chemical substance changes, certain characteristic peaks in the Raman spectrum will either increase or decrease. Therefore, by analyzing the changes in Raman signals at different solution concentrations, it is possible to identify samples with varying concentrations, while the signals of other substances may not exhibit significant changes. By comparing the spectra at different concentrations, one can

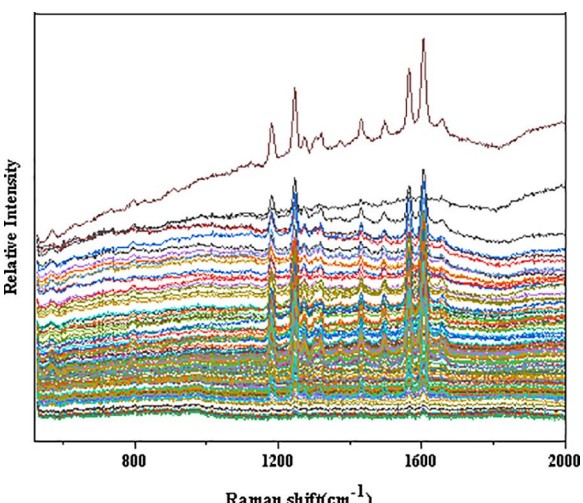

**Fig 1. Original raman data.**

clearly observe which peaks' intensities are affected by concentration changes. However, pure spectral analysis still suffers from the drawbacks of insufficient accuracy and low efficiency.

By quantitatively analyzing parameters including characteristic peak intensity, shift, and bandwidth in Raman spectra, we developed a deep learning-based concentration regression prediction model. The methodology involves: first extracting feature peak intensities (such as the C=O stretching vibration at 1600 cm$^{-1}$) as input features, then establishing concentration prediction models using algorithms like support vector regression (SVR). After training with samples of known concentrations, the model outputs continuous concentration predictions for unknown samples (with precision reaching ±0.08 µg/mL). This regression approach was selected based on Tilianin's concentration-dependent pharmacological characteristics and the strict quantitative requirements (±5% error margin) for drug content measurement specified in the Chinese Pharmacopoeia, thereby meeting pharmaceutical quality control demands for precise concentration determination.

In addition, as seen in the figure, the data baseline exhibits significant drift due to various factors, leading to large intensity deviations. The baseline correction is performed using airPLS,and after correction, the baseline of the data becomes unified, as shown in Fig 2.

To observe the peak intensity differences of solutions at various concentrations, the average Raman data of six different concentrations of Tilianin solutions were compared, as shown in Fig 3. As the Tilianin concentration in the solution increased, the intensity of several peaks significantly increased, with notable peaks around Raman shifts of 1180 cm$^{-1}$, 1245 cm$^{-1}$, 1565 cm$^{-1}$, and 1604 cm$^{-1}$.The concentration of Tilianin in the solution directly affects the intensity of the Raman signal. Specifically, as the concentration of the solution increases, the number of Tilianin molecules in the solution increases, meaning that more Tilianin molecules interact with the laser beam per unit volume, leading to an enhancement of the Raman scattering signal. By monitoring the changes in the intensity of specific Raman peaks (such as those associated with C-H, C=C, C-O, etc.), the concentration variation of Tilianin can be analyzed. The higher the concentration, the greater the intensity of these characteristic peaks, and conversely, the intensity is weaker at lower concentrations. By combining the molecular structure of Tilianin, these peaks can be identified.

As Table 1 shows, the Raman peak of C-O in Tilianin solution typically appears around 1000 cm$^{-1}$. As the concentration increases, the intensity of this peak also increases. By establishing a relationship between concentration and Raman signal intensity, quantitative analysis can be performed. The peak at 1180 cm$^{-1}$ may be related to the stretching vibration of C-H bonds in the molecule, while the peak at 1245 cm$^{-1}$ represents characteristic peaks of the C-H bending vibration

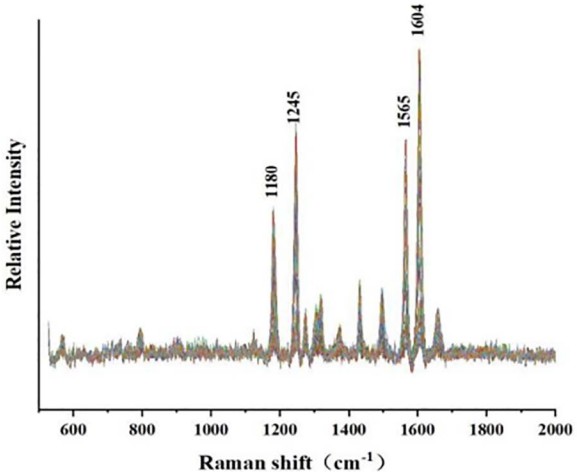

**Fig 2. Preprocessed Raman Spectra of Tilianin (airPLS baseline correction, SG smoothing, and normalization.**

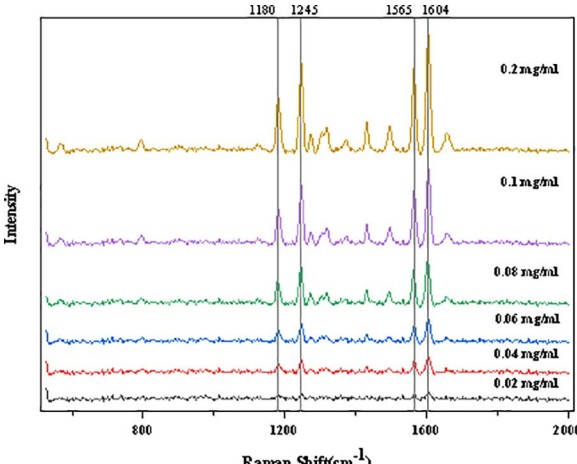

**Fig 3. The average Raman data of six solution concentrations.**

**Table 1. Reference table assigning various Raman peaks.**

| Wavenumber (cm⁻¹) | Corresponding substance | RS |
|---|---|---|
| 1180 | stretching vibration of C-H bonds | √ |
| 1245 | C-H bending vibration<br>C=C stretching vibration on the benzene ring | √ |
| 1565 | stretching vibration of C-H bonds on the aromatic ring | √ |
| 1604 | C C in-plane bending mode of phenylalanine and tyrosine | √ |

and the C=C stretching vibration on the benzene ring. The peak at 1565 cm⁻¹ is due to the stretching vibration of C-H bonds on the aromatic ring, and the peak at 1604 cm⁻¹ corresponds to the C=C stretching vibration on the phenyl group. These can all be correlated with the molecular structure of Tilianin [27–28]. Additionally, as shown in the Fig 3, it is clearly observed that as the concentration decreases, the spectral signal of the Tilianin solution at 0.02 mg/ml becomes relatively weak, and the characteristic peaks are no longer prominent. Therefore, the detection concentration threshold for Tilianin solution is determined to be 0.02 mg/mL in this experiment.

## 3. Data modeling evaluation and application

### 3.1. Model construction

The deep learning model constructed in this paper can achieve high-accuracy quantitative analysis of Tilianin, using convolutional layers to extract local features from the spectrum.

The convolutional layers and pooling operations hierarchically compress the feature space while retaining important feature distributions. Subsequently, $1 \times 1$ convolutional channel compression is applied to reduce the channel dimensions and strengthen local features. The $1 \times 1$ convolutional channel compression reduces the 256-dimensional features to 64 dimensions through cross-channel linear combination (with ReLU activation). The computational formula is expressed as:

$$Z_{out} = ReLU(w_{64 \times 256} * Z_{in} + b) \tag{1}$$

Since CNNs are adept at extracting local features, which are suitable for information such as local peaks and valleys in spectral data, the features extracted by the CNN are concatenated with the original input features to form a rich,

multi-level representation. These are then fused through fully connected layers to perform feature fusion and regression prediction.

In this section, we address the vanishing gradient problem and accelerate model convergence by introducing short-cut paths based on residual learning principles. Specifically, the input features are directly added to transformed outputs through a series of operations that typically include: batch normalization (for stabilizing intermediate feature distributions), ReLU activation functions (introducing nonlinearity), and convolutional layers (for feature extraction). This design improves effective gradient propagation, enhances the model's representational capacity, and enables the training of deeper networks. Traditional regression models demonstrate limited capability in extracting and modeling high-dimensional complex feature data, failing to fully utilize the nonlinear characteristics of spectral data. As continuous one-dimensional signals, spectral data exhibit significant local patterns and global trends, making them particularly suitable for joint processing through convolutional neural networks and attention mechanisms.

Improving prediction accuracy and model generalization capability constitutes the core objective of this experiment. The hybrid module design effectively extracts multi-level features through synergistic operations: while the self-attention module demonstrates powerful global feature capture ability, this architectural combination enhances the model's capacity for multi-scale feature representation – convolutional layers extract local spectral details, whereas the self-attention mechanism learns long-range dependencies across the entire spectrum.Spectral data inherently exhibit both local continuity and global trends, making them particularly amenable to convolutional and attention-based processing respectively. As illustrated in Fig 4, the model architecture combining self-attention mechanisms with fully connected layers employs convolutional layers to progressively extract deep features from input data. Specifically, 1 × 1 convolutional kernels perform linear transformations across channels to enable cross-channel information exchange. The processed data is then concatenated with the original input, allowing simultaneous capture of both shallow and deep features. Furthermore, each convolutional layer is followed by max pooling with a window size of 2 to enhance computational efficiency. The model employs Rectified Linear Units (ReLU) as activation functions to enhance network expressiveness, with Mean Squared Error (MSE) serving as the loss function. Adaptive learning rate adjustment is achieved through Stochastic Gradient Descent (SGD) with L2 regularization, simultaneously accelerating training convergence while preventing overfitting. Experimental parameters were set as follows: initial learning rate = 0.01, batch size = 32, iterations = 100 epochs, and dropout rate = 0.5 after fully connected layers.All models in this study were implemented using PyTorch 1.12.1 framework. This meticulously designed architecture preserves local spectral feature details while capturing long-range dependencies through global attention mechanisms, providing a reliable solution for Tilianin concentration prediction.

To enable the model to refine critical information while considering global context, the self-attention mechanism performs additional feature extraction on the data. The processed features are then flattened and fed into a fully-connected network for concentration prediction, employing a three-layer architecture.The incorporation of self-attention mechanisms significantly enhances the model's capability to represent Raman spectral features. As illustrated in Fig 5, the self-attention mechanism enables the model to learn interdependencies across all wavelengths rather than just local patterns.

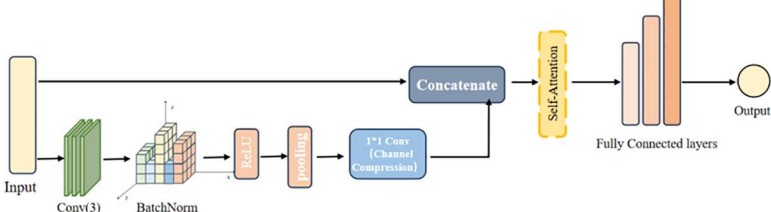

**Fig 4. The structure of RSAQN.**

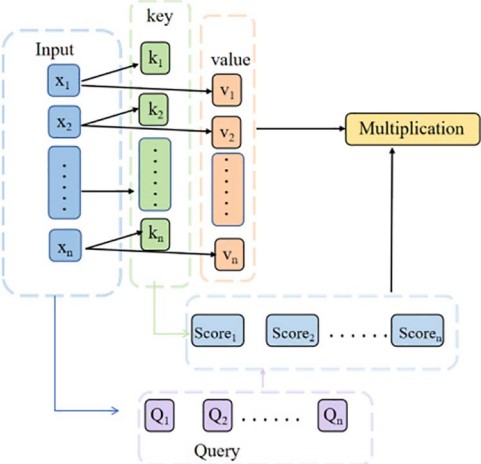

**Fig 5. Self-Attention for Raman Spectroscopy feature extraction.**

This proves particularly crucial for complex spectral data analysis because Raman spectral correlations often extend beyond adjacent wavelengths.Finally, the self-attention mechanism demonstrates dynamic wavelength focusing capability during training, enabling the model to autonomously learn and prioritize the most spectrally relevant features for specific analytical tasks (e.g., component quantification, material identification).

During model training, we utilized 20 spectral measurements per concentration level, with a 70:30 split between training and test sets respectively. To mitigate overfitting given the limited dataset size, we implemented 5-fold cross-validation for partitioning the training and validation sets.

### 3.2. Model Evaluation and Application.

To evaluate the performance of the model, we employed several common regression evaluation metrics, including Mean Squared Error (MSE), Mean Absolute Error (MAE), and $R^2$ (coefficient of determination). These metrics provide a comprehensive reflection of the model's accuracy and stability in the prediction task. MSE measures the sum of the squared deviations between predicted and actual values, while MAE focuses on the absolute value of prediction errors; both effectively assess the model's performance on the training and test sets. $R^2$ reflects the model's ability to explain the data, with values closer to 1 indicating a better fit.Additionally, we used cross-validation to further evaluate the model's generalization ability. By splitting the dataset into multiple subsets for training and validation, cross-validation helps effectively prevent overfitting, ensuring the model's stability and reliability on new data.

To achieve better stability and prevent overfitting, various network architectures and parameter settings were tested during the model construction process. The primary comparisons of model architectures are shown in Table 2.During the model construction process, ablation experiments were conducted to compare various network architectures and parameter configurations. The purpose of these ablation comparisons was not only to observe improvements in model performance but, more importantly, to analyze the mechanisms behind different design choices in deep learning models and identify which components and parameters significantly impact the final performance. L2 regularization was applied by adding the square of the weights to the loss function, encouraging the network weights to remain small, thereby mitigating overfitting to some extent. This weight decay mechanism limits the magnitude of the weights during training, reducing the model's tendency to overfit the training data. By introducing L2 regularization, the experimental results showed that the $R^2$ value increased by over 10%, from 0.78, and the error was significantly reduced. This indicates that L2 regularization effectively enhanced the model's generalization ability, making the predictions more stable and accurate.

**Table 2. The impact of different model parameters on performance.**

|  | $R^2$ | MSE | RMSE | MAE |
|---|---|---|---|---|
| **Without L2** | 0.7803 | 0.0007 | 0.0278 | 0.0250 |
| **With Adam optimizer** | 0.9000 | 0.0003 | 0.0188 | 0.0152 |
| **With Random partitioning** | [0.69-0.91] | – | – | – |
| **Without self-Attention** | 0.9031 | 0.0003 | 0.0185 | 0.0141 |
| **RSAQN** | 0.9144 | 0.0003 | 0.0173 | 0.0136 |

The Adam optimizer combines the momentum method and RMSProp, adaptively adjusting the learning rate, making it suitable for handling sparse data and situations with unstable gradients. In contrast, the SGD optimizer offers better convergence, particularly when the data distribution is uniform or the batch size is large. Comparing these two optimizers helps determine which performs better for processing spectral data. To identify the optimal choice, the experiment compared the performance of the regularized Adam optimizer and the regularized SGD optimizer. The results showed that the difference between the two was minimal, but SGD performed slightly better. Specifically, using SGD increased the $R^2$ value by nearly 2% compared to Adam. This improvement is because SGD is more suitable for handling uniformly distributed data and can achieve better convergence when an appropriate learning rate is used.

During the model construction process, prediction instability was observed, with the $R^2$ values fluctuating between 0.69 and 0.91. This instability was likely due to the small dataset size or uneven sample distribution. To address this, the experiment divided the training and test sets according to six concentration levels in specific proportions, rather than using a random split of the entire dataset.The results showed that this approach improved model stability, reducing the $R^2$ fluctuation range to between 0.91 and 0.92. This not only increased the $R^2$ value but also reduced the prediction error. The instability made it difficult to obtain reliable statistical results for other metrics within a stable range. Therefore, the full dataset for the "With Random Partitioning" row is not presented.

Ablation experiments were also conducted on the self-attention mechanism to validate its role in the model. The results demonstrated that self-attention improves model performance by focusing on key information. The self-attention mechanism dynamically adjusts attention weights during feature extraction, enabling the model to concentrate on important features and critical information. This is particularly significant for capturing specific peaks and patterns in spectral data. Through these ablation experiments and analyses, the rationale behind each module and parameter selection was comprehensively verified. This meticulous experimental design not only enhanced the model's accuracy and stability but also provided an optimized technical approach for spectral data analysis, laying a solid foundation for practical applications.

### 3.3. Model comparison experiments

For comparative analysis, this study selected traditional machine learning models including Random Forest (RF) and K-Nearest Neighbors (KNN), along with deep learning approaches such as Convolutional Neural Networks (CNN) and Variational Autoencoders (VAE) [29] as benchmarks. All models were processed with identical preprocessing steps, with their performance comparisons detailed in Table 3.The Random Forest (RF) model was configured with 100 decision trees, where each tree's maximum depth was limited to 20 layers to prevent overfitting. During node splitting, multiple features were considered, with a minimum leaf sample size set to 5. The model achieved a final $R^2$ of 0.6498, with its performance limitations likely attributable to insufficient data volume and the model's inherent constraints in continuous value prediction.The model achieved $R^2 = 0.8443$, outperforming Random Forest (RF) but remaining inferior to deep learning approaches. The Artificial Neural Network (ANN) adopted a three-layer fully-connected architecture (128-64-1 neurons), with ReLU activation in hidden layers and linear activation in the output layer. Trained using the Adam optimizer (learning rate = 0.001), it attained $R^2 = 0.8974$, demonstrating the potential of traditional neural networks for small-sample problems.

**Table 3. Comparison of performance with traditional machine learning and other deep learning models.**

|  | R² | MSE | RMSE | MAE |
|---|---|---|---|---|
| **RF** | 0.6498 | 0.0012 | 0.0351 | 0.0318 |
| **KNN** | 0.8443 | 0.0005 | 0.0234 | 0.0212 |
| **ANN** | 0.8974 | 0.0003 | 0.0190 | 0.0147 |
| **CNN** | 0.8432 | 0.0005 | 0.0230 | 0.0187 |
| **VAE** | 0.8266 | 0.0005 | 0.0242 | 0.0178 |
| **RSAQN** | 0.9144 | 0.0003 | 0.0173 | 0.0136 |

The comparative VAE model employed a two-layer 3×1 convolutional encoder (32→64 channels) to compress inputs into an 8-dimensional latent space (β=0.5), with the decoder reconstructing spectra through fully connected layers and transposed convolutions. The model achieved a prediction R² of 0.8266, with its performance limitations likely attributable to information loss in the latent space compression.

The proposed model in this study outperformed the other five comparison models, achieving an R² value of 0.9144. In the RF experiments, the number of trees was set to 10, 50, and 100 respectively, with the optimal result selected as RF's performance metric. The results showed its R² reached only 0.6498, with errors 2–4 times higher than those of the proposed model, which may be attributed to insufficient data volume or inherent limitations of the model itself.The KNN model predicts target values based on neighboring data points, demonstrating superior performance to RF with an R² of 0.8443. The ANN, implemented as a three-layer neural network, achieved an R² of 0.8974, showing promising potential for small-scale problems while still underperforming compared to the deep learning model developed in this study.

The architectures of the two comparative deep learning models are shown in Fig 6. The CNN model consists of two convolutional layers and two fully connected layers. The VAE uses convolutional layers and transposed convolutional layers as its encoder and decoder respectively, with two fully connected layers outputting the prediction results. The VAE decoder does not reconstruct the original spectra at its output end, but directly predicts concentration values. Since drug concentration prediction is a regression task, directly mapping the corresponding concentration values in the latent space can avoid information loss. After encoding the local characteristic peaks of Raman spectra in the latent space, gradient backpropagation verification confirms that key quantitative information remains preserved.The performance of both comparative deep learning models was comparable to KNN but significantly inferior to the proposed model in this study. This discrepancy stems from our model's architecture, which preserves original features through residual connections,

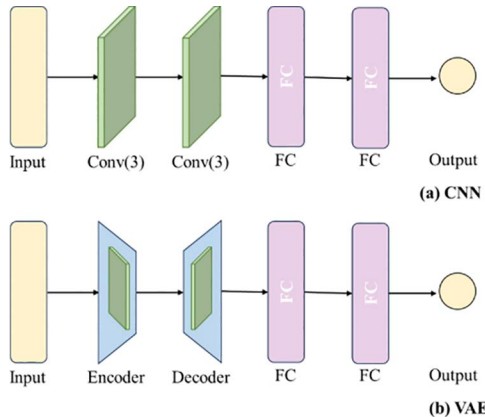

**Fig 6. Comparative deep learning models (a)CNN (b)VAE.**

enabling smoother and more stable learning that enhances both accuracy and generalization capability. In contrast, CNN and VAE may lose partial original features during their encoding processes.

Overall, compared to traditional machine learning, the structure and parameter adjustability of deep learning models support continuous optimization and personalized design in specific domains, providing strong support for future quantitative analysis research. Experimental validation shows that the combination of residual structures and self-attention mechanisms used in this study outperforms commonly used CNN and VAE architectures.

### 3.4. Model testing and analysis

The model's predictive performance for six Tilianin concentrations on the test set is shown in Fig 7. In the low concentration range of 0.02 mg/ml, the model demonstrates stable prediction results with fluctuations controlled within 0.02, indicating excellent quantitative accuracy and robustness at low concentrations. This capability holds significant importance for practical applications in quality control and monitoring of low-concentration Tilianin, particularly for clinical efficacy evaluation and pharmacokinetic studies where accurate tracking of low drug concentration levels in vivo is crucial.At the concentration of 0.04 mg/ml, the characteristic Raman spectral peaks gradually intensify, with signal strength demonstrating stable linear growth. The vibrational modes of Tilianin's key functional groups become more distinct, providing enhanced signal support for quantitative analysis.

At concentrations of 0.06 mg/ml, 0.08 mg/ml, and 0.1 mg/ml, the intensity of Raman characteristic peaks increases significantly, though signal saturation or nonlinear variations may occur. This suggests enhanced intermolecular interactions among Tilianin molecules at higher concentrations, potentially leading to spectral changes such as peak broadening or shifts. In the high concentration range of 0.2 mg/ml required for tumor therapy, quantitative analysis ensures drug concentrations remain within safe and effective ranges while detecting the effects of high-concentration Tilianin on cells or tissues, providing evidence for defining safe dosage limits.

However, solvent-induced baseline drift cannot be adaptively corrected by linear models like PLS, while traditional deep learning models are prone to overfitting and fail to effectively extract features required for quantitative analysis. Our designed RSAQN model utilizes residual connections to preserve original peak positions, employs multi-scale convolutional kernels to capture characteristic peaks, and enhances model expressiveness through attention mechanisms, enabling the training of deeper networks. Traditional regression models have limited capability in extracting and modeling high-dimensional complex feature data, being unable to fully leverage the nonlinear characteristics of spectral data. As

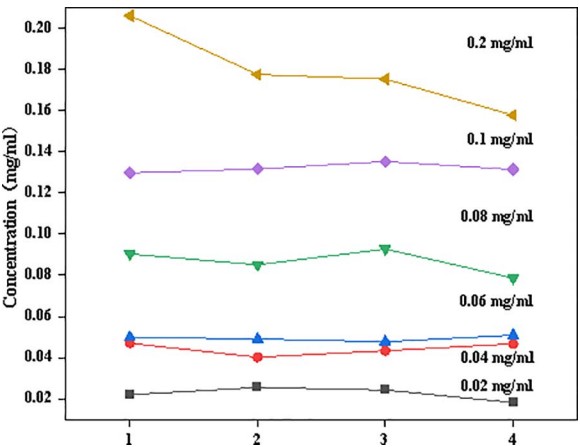

**Fig 7. Model's prediction results on the test set.**

continuous one-dimensional signals, spectral data exhibit distinct local patterns and global trends, making them particularly suitable for combined processing via convolutional neural networks and attention mechanisms. Furthermore, our model specifically addresses small-sample optimization design challenges, significantly improving generalization capability under data scarcity conditions.

To investigate the performance of SGD (Stochastic Gradient Descent) and Adam (Adaptive Moment Estimation) optimizers under different configurations in this experiment, we systematically adjusted learning rates and batch sizes while conducting multiple experimental trials. Tables 4 and 5 present detailed analyses and validation results comparing the optimizers' performance.

The experimental comparison results demonstrate that Adam's adaptive learning rates for each parameter typically enable faster convergence during the initial training phase. In contrast, SGD employs a fixed learning rate – excessively high values may cause training instability, while overly low values can lead to slower training progress.

SGD typically employs smaller batch sizes, introducing noise in each parameter update that increases training stochasticity, potentially helping escape local optima. In contrast, Adam generally uses larger batches, resulting in more precise parameter updates.In terms of batch size settings, SGD performs better with a batch size of 32, but its performance slightly declines when the batch size increases to 64. In contrast, Adam shows relatively poorer performance with larger batch sizes (e.g., 64), likely because the larger batch size leads to smoother optimization trajectories, making Adam more prone to getting trapped in local optima during solution space exploration.

To further validate the prediction accuracy and robustness of the proposed model in the low concentration range, two additional supplementary experiments of 5-fold cross-validation and noise injection test were introduced in this study, as shown in Table 6. Firstly, the generalization ability of the model under different data split was evaluated through 5-Fold Cross Validation. The results showed that the model maintained high consistency on multiple sub-datasets, obtaining $R^2 = 0.911497$ and RMSE $= 0.0195$. It indicates that it has good stability. Second, to evaluate the robustness of the model

**Table 4. Comparison of SGD and adam with different learning rates.**

| Optimizer | Learning Rate | Batch Size | R² | RMSE | MAE | MSE |
|---|---|---|---|---|---|---|
| SGD | 0.01 | 32 | 0.91 | 0.0003 | 0.0173 | 0.00036 |
| SGD | 0.001 | 32 | 0.85 | 0.0225 | 0.0160 | 0.00040 |
| Adam | 0.01 | 32 | 0.87 | 0.0205 | 0.0152 | 0.00037 |
| Adam | 0.001 | 32 | 0.82 | 0.0243 | 0.0183 | 0.00045 |

**Table 5. Batch size comparison.**

| Optimizer | Learning Rate | Batch Size | R² | RMSE | MAE | MSE |
|---|---|---|---|---|---|---|
| SGD | 0.01 | 32 | 0.91 | 0.0003 | 0.0173 | 0.00036 |
| SGD | 0.01 | 64 | 0.87 | 0.0212 | 0.0165 | 0.00040 |
| Adam | 0.01 | 32 | 0.87 | 0.0205 | 0.0152 | 0.00037 |
| Adam | 0.01 | 64 | 0.85 | 0.0221 | 0.0173 | 0.00041 |

**Table 6. Experimental results of cross-validation and noise injection.**

| Verification method | MSE | RMSE | MAE | R² |
|---|---|---|---|---|
| cross-validation | 0.000380 | 0.0195 | 0.0151 | 0.911497 |
| noise injection | 0.000391 | 0.0198 | 0.0155 | 0.887597 |
| Final test | 0.000382 | 0.019546 | 0.015075 | 0.887269 |

under data perturbation, we applied Gaussian noise (mean 0, standard deviation 0.05) to the training set for noise injection testing.The test results show that although the error increases slightly ($R^2 = 0.887597$), it is still within the acceptable range, verifying that the model can still maintain good performance when facing certain background noise in the experimental environment or practical applications.

Finally, we conducted the final performance evaluation of the model on the retained test set and obtained $R^2 = 0.887269$. All the indicators were basically consistent with the results of cross-validation and noise tests, further confirming the accuracy and robustness of the proposed model.

As shown in Fig 7, the prediction curves for five concentration levels (0.02 mg/ml to 0.1 mg/ml) exhibit relatively smooth trends with fluctuations controlled within 0.02, while the 0.2 mg/ml solution predictions demonstrate significant variations. Overall, this evaluation provides essential data support for formulation development by characterizing Tilianin's solubility properties across different solvents or dosage forms.The quantitative analysis results of Tilianin at different concentrations correspond to specific practical applications: low concentrations suit trace detection and early efficacy monitoring, medium concentrations are ideal for standardized analysis and quality control, while high concentrations serve therapeutic monitoring and toxicology studies. These quantitative outcomes not only support pharmaceutical quality control but also play vital roles in clinical applications, personalized medicine, and pharmacokinetic research.

We calculated the deviation rates between the predicted values and standard reference values for each concentration sample, along with their corresponding deviations. Under consistent model training and testing conditions, the results presented in Table 7 demonstrate that all deviation rates were maintained within 10%.

## 4. Discussion

The application of Raman spectroscopy combined with deep learning technology in pharmaceutical analysis offers significant advantages, primarily in terms of high sensitivity, non-invasiveness, speed, and intelligent data analysis.Raman spectroscopy, by detecting molecular vibration information, can accurately reflect the chemical structure and concentration changes of drug molecules, making it particularly suitable for analyzing mixed samples with multiple components in complex systems.Traditional methods, such as High-Performance Liquid Chromatography (HPLC) or Mass Spectrometry (MS), though accurate, often involve complex procedures, high costs, and slower analysis speeds. Deep learning, as a powerful data modeling tool, can automatically extract deep features from spectral data, overcoming the limitations of traditional statistical methods in processing high-dimensional data. The model effectively captures key features in spectral data and enhances its generalization capability through the incorporation of residual connections and self-attention mechanismsRashedi et al. [30] proposed integrating variational autoencoders (VAE) with just-in-time learning for Raman spectroscopy-based cell culture process monitoring, enhancing the model's adaptability and feature representation capability. Min et al. [31] developed a semi-supervised convolutional neural network (CNN) regression model incorporating data augmentation and spectral annotation, significantly improving Raman modeling performance for cell culture prediction, particularly in label-scarce scenarios. Khodabandehlou et al. [32] developed a convolutional neural network (CNN)-Raman spectroscopy hybrid method for predicting product quality attributes during cell culture processes, achieving high-precision quality prediction. While these studies primarily focus on biopharmaceutical cell culture monitoring (typically relying on large-scale process Raman data), our research specifically addresses the quantitative analysis of natural drug active component Tilianin, particularly developing models under small-sample and powdered-sample conditions. This application scenario remains relatively unexplored in existing literature and presents challenges including subtle spectral

**Table 7. Deviation rate of predicted values.**

| Concentration (mg/mL) | 0.2 | 0.1 | 0.08 | 0.06 | 0.04 | 0.02 |
|---|---|---|---|---|---|---|
| Average Predicted Values (mg/mL) | 0.179 | 0.091 | 0.075 | 0.059 | 0.037 | 0.018 |
| Deviation Rate | 10% | 9% | 6% | 2% | 7% | 1% |

variations and limited data availability. Compared to previous approaches employing complex architectures like variational autoencoders (VAE) and semi-supervised CNNs, our proposed custom network RSAQN offers lightweight structure and stable performance, making it particularly suitable for drug detection tasks with constrained data. Furthermore, we emphasize model interpretability – a critical requirement in practical pharmaceutical analysis.

## 5. Conclusion

In this study, the quantitative analysis method for Tilianin based on Raman spectroscopy and a one-dimensional deep learning model not only achieved high-accuracy content prediction ($R^2 = 0.9144$), but also demonstrated superior predictive performance compared to traditional models such as Random Forest (RF), K-Nearest Neighbors (KNN), Artificial Neural Networks (ANN), Convolutional Neural Networks (CNN), and Variational Autoencoders (VAE).This indicates that Raman spectroscopy combined with deep learning technology can quickly and accurately predict drug content, providing an efficient and convenient solution for drug development and quality control. It also offers a reference for the rapid quantitative analysis of other bioactive compounds, further expanding the potential applications of Raman spectroscopy in pharmaceutical analysis.This provides an efficient detection method for determining and measuring therapeutic drugs used to inhibit tumor cell proliferation, induce apoptosis, and exert antitumor effects. These results are expected to serve as a reference for future clinical testing, enabling more accurate qualitative determination of drug concentrations in the serum of cancer patients through Raman spectroscopy diagnostic methods and classification models. This advancement aims to develop a rapid, sensitive, and highly reliable Raman spectroscopy detection system, driven by spectral diagnostic technology and artificial intelligence, to assist medical practitioners in formulating personalized treatment strategies for patients.

## Author contributions

**Conceptualization:** Wen Jiang, Xiaomei Pan.

**Data curation:** Jieyu Liu, Yanqi Ma.

**Formal analysis:** Wen Jiang, Yanqi Ma.

**Funding acquisition:** Xiaotong Xin.

**Investigation:** Xiaotong Xin, Junhui Chen.

**Project administration:** Junhui Chen.

**Resources:** Jieyu Liu.

**Software:** Wei Zhang, Jieyu Liu, Cheng Chen.

**Supervision:** Wei Zhang, Cheng Chen.

**Writing – original draft:** Wei Liu, Jieyu Liu, Xiaomei Pan.

**Writing – review & editing:** Wei Liu.

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
