## [Decision Letter · Decision Letter 0]

Dear Dr. Pan,

Thank you for submitting your manuscript to PLOS ONE. After careful consideration, we feel that it has merit but does not fully meet PLOS ONE’s publication criteria as it currently stands. Therefore, we invite you to submit a revised version of the manuscript that addresses the points raised during the review process.

We look forward to receiving your revised manuscript.

Kind regards,

Clara Sousa

Academic Editor

PLOS ONE

 [This work was sponsored by Natural Science Foundation of Xinjiang Uygur AutonomousRegion((2023D01C158), the "Tianshan Talents" Medical and Health High Level Talent Training Program Project (TSYC202301B068), and the Natural Science Foundation of Xinjiang Uygur Autonomous Region - General Project (2024D01C157)]. 

[This work was sponsored by Natural Science Foundation of Xinjiang Uygur AutonomousRegion(2023D01C158), the "Tianshan Talents" Medical and Health High Level Talent Training Program Project (TSYC202301B068), and the Natural Science Foundation of Xinjiang Uygur Autonomous Region - General Project (2024D01C157)]

[This work was sponsored by Natural Science Foundation of Xinjiang Uygur AutonomousRegion((2023D01C158), the "Tianshan Talents" Medical and Health High Level Talent Training Program Project (TSYC202301B068), and the Natural Science Foundation of Xinjiang Uygur Autonomous Region - General Project (2024D01C157)]. 

6. We note that you have indicated that there are restrictions to data sharing for this study. For studies involving human research participant data or other sensitive data, we encourage authors to share de-identified or anonymized data. However, when data cannot be publicly shared for ethical reasons, we allow authors to make their data sets available upon request. For information on unacceptable data access restrictions, please see http://journals.plos.org/plosone/s/data-availability#loc-unacceptable-data-access-restrictions.

7. In the online submission form, you indicated that [The datasets generated and analyzed during the current study are not publicly available, but are available from the corresponding author upon reasonable request.].

8. Please include a copy of Table 2 which you refer to in your text on page 17.

Reviewers' comments:

Reviewer's Responses to Questions

**Comments to the Author**

1. Is the manuscript technically sound, and do the data support the conclusions?

Reviewer #1: Yes

Reviewer #2: Partly

2. Has the statistical analysis been performed appropriately and rigorously?

Reviewer #1: Yes

Reviewer #2: I Don't Know

3. Have the authors made all data underlying the findings in their manuscript fully available?

Reviewer #1: No

Reviewer #2: No

4. Is the manuscript presented in an intelligible fashion and written in standard English?

Reviewer #1: No

Reviewer #2: No

Reviewer #1: The comments are attached as a pdf file, but the PLOS One website cannot accept the attachment only. So, I include all the comments so that I can proceed with submission. As a result, I request the authors to refer to the attachment to have a better understanding of the comments.

The manuscript presents an innovative approach to the quantitative analysis of Tilianin using Raman spectroscopy combined with deep learning. This work proposes an approach for Tilianin quantification and seems likely to compare the proposed deep learning model with other traditional and machine learning models; however, there are some weaknesses that should be addressed before being considered for publication.

The readability of the manuscript is rather low. Particularly, the introduction is poorly written. Please take a careful consideration to revise the text to improve the readability.

The authors are requested to cross check the references to ensure they are placed properly. For instance, in Introduction section, the reference 22 discusses CNN and not VAE. Please verify.

The last paragraph of Introduction is poorly written, and the novelties are very unclear. The authors need to spend some time to improve the quality of writing and describe all the contributions of the work clearly.

There are other works in the literature that have addressed similar challenges with either a greater number of data or data augmentation techniques. Some of these works are:

Rashedi, M., Khodabandehlou, H., Wang, T., Demers, M., Tulsyan, A., Garvin, C., and Undey, C. (2024). Integration of just-in-time learning with variational autoencoder for cell culture process monitoring based on Raman spectroscopy. Biotechnol. Bioeng, 121, 1–20. doi:10.1002/bit.28713

Min, R., Wang, Z., Zhuang, Y., and Yi, X. (2023). Application of semi-supervised convolutional neural network regression model based on data augmentation and process spectral labeling in Raman predictive modeling of cell culture processes. Biochemical Engineering Journal, 191(108774), ISSN 1369–703X.

Khodabandehlou, H., Rashedi, M., Wang, T., Tulsyan, A., Schorner, G., Garvin, C., and Undey, C. (2024). Cell culture product quality attribute prediction using convolutional neural networks and raman spectroscopy. Biotechnol. Bioeng, 121, 1231–1243. doi:10.1002/bit.28646.

The authors are requested to describe the contributions of their work compared to the mentioned references.

In many places in the text, the word “And” is used right after the end of a finished sentence. Please correct/remove them throughout the manuscript.

In section 2.1 it is written “Weigh the desired mass of Tilianin powder on a balance and transfer it to a beaker. Using a pipette measure less than the target volume of methanol into the beaker. Heat gently and stir well using a glass rod until Tilianin is completely dissolved. Transfer the dissolved Tilianin solution to a volumetric flask. Use a standard volumetric flask to finalize the volume and rinse the residual material several times to ensure that all Tilianin enters the solution system.” This type of writing is not a common practice in academic writing and needs a major revision. Please revise these sentences throughout the text accordingly.

In section 2.2, the authors are needed to explain how the offline measurements are labeled with the Raman spectra in the training dataset.

Section 2.3, de-meaned � detrended

In section 2.3 appropriate references are required for the baseline removal. Some of those references are:

Eilers, P., & Boelens, H. F. M. (2005). Baseline correction with asymmetric least squares smoothing (Leiden University Medical Centre Report).

Cai, Y., Yang, C., Xu, D., & Gui, W. (2018). Baseline correction for Raman spectra using penalized spline smoothing based on vector transformation. Analytical Methods, 10, 3525–3533.

Please consider providing comparison with these references to validate your approach.

In figure 1, the wave numbers are shown from 550 cm^(-1) to 2000 cm^(-1). Is this the ultimate range of wavenumbers? What happens to the spectra before or after this range? How many wavenumbers does the Raman machine provide?

In second paragraph of section 2.4, the authors have mentioned a classification model can be made between Raman spectra and concentrations. Do the authors mean a regression model? Otherwise, please clearly explain what the classes are in the classification. Then, please explain why classification is needed?

In the caption of figure 2, please type in what preprocessing steps are implemented on the Raman spectra.

In section 3.1, what do the authors mean by 1×1 convolutional channel compression? Please clarify. Also, please show where this is applied on Figure 4.

In the third paragraph of section 3.1, the authors discuss residual blocks. They say “the input features are directly added to a series of transformed outputs”. They need to clarify which output they are talking about. I don’t see any residual block for the output of the model in Figure 4.

The authors have tried to report the hyper parameters of the model structure in section 3.1, however, there are some missing details like the structure of fully connected layer.

In section 3.3 where the models are compared, a detailed description of all models is required. The competitive models lack detailed information regarding their hyper-parameters. Please revise this section appropriately.

Is this VAE model generated by the authors or has been used in other works? There are some curiosities about this model since a regenerated input in the decoder is used in fully connected layers. This provides a question why a regeneration of input improves the performance of compared with regular ANN. Please comment on this and clarify.

The authors are requested to discuss what specific challenges in Raman spectral analysis the proposed model is designed to overcome compared to prior models.

Why was SGD found to perform better than Adam? Could this be due to batch size or learning rate settings?

The self-attention mechanism is mentioned but not well explained in the context of spectral analysis. A brief mathematical or visual representation of how self-attention aids in feature extraction from Raman spectra would be beneficial.

The model performs well in lower concentration ranges but exhibits fluctuations at 0.2 mg/ml. What might be causing this? Is this due to saturation effects in Raman spectral intensities, or does the deep learning model struggle with higher concentration ranges?

While the paper shows strong performance metrics, a discussion on how RSAQN could be integrated into real pharmaceutical quality control workflows would be useful. Were any misclassification or prediction errors analyzed qualitatively? What types of spectra did the model struggle with?

Consider discussing potential applications beyond Tilianin (e.g., broader pharmaceutical compounds, food safety, biomedical diagnostics). Could transfer learning be applied to enhance model generalization across different Raman spectroscopy datasets?

Reviewer #2: This study presents a novel Raman-based deep learning model (RSAQN) for non-destructive Tilianin quantification. In contrast to conventional destructive HPLC/MS techniques, RSAQN employs self-attention mechanisms to analyze spectral features from six concentration levels (120 samples total). The model demonstrates superior performance over five benchmark ML/DL approaches, achieving an R² value of 0.9144, thereby enabling rapid and accurate drug quality assessment. However, several issues require attention:

1. The current English expression needs improvement - the descriptions are unnecessarily complex, sentences are overly long, and the overall organization lacks clarity. A more concise and straightforward presentation would significantly enhance readability.

2. The Introduction section disproportionately focuses on HPLC, which is not the manuscript's primary focus. The HPLC discussion should be condensed, while more emphasis should be placed on Tilianin measurement techniques and relevant machine learning methods. A comparative analysis of key parameters would better highlight the advantages of the proposed method.

3. In the Data Acquisition section: The characteristic peak at 520.7 cm-1 - which molecular component does this correspond to?

4. For the Spectral Data Analysis section: Including a reference table assigning various Raman peaks would significantly improve data interpretation.

5. In the Model Testing and Analysis section: The meaning of "fluctuations controlled within 0.02" is unclear - does this refer to an absolute value of 0.02 or 2% variation? Additionally, the claim regarding the model's accuracy and robustness in low-concentration ranges (based solely on Figure 2) requires stronger validation. Additional methods such as cross-validation and noise injection testing should be incorporated to substantiate these conclusions.

**Do you want your identity to be public for this peer review?** For information about this choice, including consent withdrawal, please see our Privacy Policy

Reviewer #1: No

Reviewer #2: No

---

## [Author Response · Author response to Decision Letter 1]

26 Apr 2025

Response Letter

Firstly, we would like to thank you for your correspondence and the constructive feedback on our article. (Study on the quantitative analysis of Tilianin based on Raman spectroscopy combined with deep learning). These comments have been immensely valuable and have greatly assisted us in improving our article. All authors have thoroughly discussed these comments. In response to the editor's suggestions, we have made every effort to revise our manuscript to meet the requirements of your esteemed journal. In this revised version, changes to our manuscript have been highlighted in red text. Below is a point-by-point response to the editor and reviewers comments.

Response 1:

We have ensured that our manuscript meets PLOS ONE's style requirements, including those for file naming.

2.Please note that PLOS ONE has specific guidelines on code sharing for submissions in which author-generated code underpins the findings in the manuscript. In these cases, we expect all author-generated code to be made available without restrictions upon publication of the work. Please review our guidelines at https://journals.plos.org/plosone/s/materials-and-software-sharing#loc-sharing-code and ensure that your code is shared in a way that follows best practice and facilitates reproducibility and reuse.

Response 2:

Our research cannot share code directly for legal or ethical reasons. We have stated this in our Data Availability Statement and can be contacted to request access to the code if needed.

3.We note that the grant information you provided in the ‘Funding Information’ and ‘Financial Disclosure’ sections do not match.When you resubmit, please ensure that you provide the correct grant numbers for the awards you received for your study in the ‘Funding Information’ section.

Response 3:

Our study was supported by [This work was sponsored by Natural Science Foundation of Xinjiang Uygur AutonomousRegion((2023D01C158), the "Tianshan Talents" Medical and Health High Level Talent Training Program Project (TSYC202301B068), and the Natural Science Foundation of Xinjiang Uygur Autonomous Region - General Project (2024D01C157)]. However, the "Tianshan Talents" Medical and Health High Level Talent Training Program Project (TSYC202301B068) is not normally retrieved and saved on the submission system, so it can only be shown in the reply letter.

4. Thank you for stating the following financial disclosure:[This work was sponsored by Natural Science Foundation of Xinjiang Uygur AutonomousRegion((2023D01C158), the "Tianshan Talents" Medical and Health High Level Talent Training Program Project (TSYC202301B068), and the Natural Science Foundation of Xinjiang Uygur Autonomous Region - General Project (2024D01C157)].

Response 4:

The funders had no role in study design, data collection and analysis, decision to publish, or preparation of the manuscript. We have amended Role of Funder statement in our cover letter

[This work was sponsored by Natural Science Foundation of Xinjiang Uygur AutonomousRegion(2023D01C158), the "Tianshan Talents" Medical and Health High Level Talent Training Program Project (TSYC202301B068), and the Natural Science Foundation of Xinjiang Uygur Autonomous Region - General Project (2024D01C157)]

[This work was sponsored by Natural Science Foundation of Xinjiang Uygur AutonomousRegion((2023D01C158), the "Tianshan Talents" Medical and Health High Level Talent Training Program Project (TSYC202301B068), and the Natural Science Foundation of Xinjiang Uygur Autonomous Region - General Project (2024D01C157)].

Response 5:

Thank you for your suggestions to us. We have deleted the part of the text related to the funding in the manuscript and placed the funding information in the cover letter.

6. We note that you have indicated that there are restrictions to data sharing for this study. For studies involving human research participant data or other sensitive data, we encourage authors to share de-identified or anonymized data. However, when data cannot be publicly shared for ethical reasons, we allow authors to make their data sets available upon request. For information on unacceptable data access restrictions, please see http://journals.plos.org/plosone/s/data-availability#loc-unacceptable-data-access-restrictions.

Response 6:

Thank you for recognizing our research, we have uploaded our research data as supplementary information We also updated Data Availability statement in the submission form accordingly.

7. In the online submission form, you indicated that [The datasets generated and analyzed during the current study are not publicly available, but are available from the corresponding author upon reasonable request.].

Response 7:

Thank you for recognizing our research, we have uploaded our research data as supplementary information We also updated Data Availability statement in the submission form accordingly.

8. Please include a copy of Table 2 which you refer to in your text on page 17.

Response 8:

We have included below a copy of Table 2, which we cite in the text on page 17.

Table 2 Comparison of performance with traditional machine learning and other deep learning models.

R2 MSE RMSE MAE

RF 0.6498 0.0012 0.0351 0.0318

KNN 0.8443 0.0005 0.0234 0.0212

ANN 0.8974 0.0003 0.0190 0.0147

CNN 0.8432 0.0005 0.0230 0.0187

VAE 0.8266 0.0005 0.0242 0.0178

RSAQN 0.9144 0.0003 0.0173 0.0136

Reviewers' comments:

Reviewer #1: The comments are attached as a pdf file, but the PLOS One website cannot accept the attachment only. So, I include all the comments so that I can proceed with submission. As a result, I request the authors to refer to the attachment to have a better understanding of the comments.

Comment 1:

The readability of the manuscript is rather low. Particularly, the introduction is poorly written. Please take a careful consideration to revise the text to improve the readability.

Response 1:

We sincerely thank the reviewers for their careful review of our introduction. We have completely rewritten the introduction section, optimizing both the article structure and language expression to significantly improve readability. All modifications have been clearly marked in red. Please refer to the revised manuscript for specific changes.

Comment 2:

The authors are requested to cross check the references to ensure they are placed properly. For instance, in Introduction section, the reference 22 discusses CNN and not VAE. Please verify.

Response 2:

We sincerely appreciate your careful review of our references. We apologize for any inaccuracies in the citations and have now thoroughly verified all references.

Comment 3:

The last paragraph of Introduction is poorly written, and the novelties are very unclear. The authors need to spend some time to improve the quality of writing and describe all the

contributions of the work clearly.

Response 3:

We sincerely appreciate your guidance regarding our introduction. Following the reviewers' comments, we have completely rewritten the concluding paragraph of the introduction to highlight the following key innovative contributions

1. Propose the first customized attention residual network (RSAQN) for Tilianin, which effectively extracts spectral features at different levels through a multi-scale convolutional neural network architecture and self attention mechanism to achieve accurate feature peak recognition;

2.Significantly improve the accuracy and sensitivity of tilianin quantitative analysis compared to traditional methods.

The modified text has been highlighted in red and kindly reviewed.

Comment 4:

There are other works in the literature that have addressed similar challenges with either a greater number of data or data augmentation techniques. Some of these works are:

a) Rashedi, M., Khodabandehlou, H., Wang, T., Demers, M., Tulsyan, A., Garvin, C., and Undey, C. (2024). Integration of just-in-time learning with variational autoencoder for cell culture process monitoring based on Raman spectroscopy. Biotechnol. Bioeng, 121, 1-20. doi:10.1002/bit.28713

b) Min, R., Wang, Z., Zhuang, Y., and Yi,X. (2023). Application of semi-supervised convolutional neural network regression model based on data augmentation and process spectral labeling in Raman predictive modeling of cell culture processes. Biochemical Engineering Journal, 191(108774), ISSN 1369-703X.

c) Khodabandehlou, H., Rashedi, M., Wang, T., Tulsyan, A., Schorner, G., Garvin, C., and Undey, C. (2024). Cell culture product quality attribute prediction using convolutional neural networks and raman spectroscopy. Biotechnol. Bioeng, 121, 1231 -1243.doi:10.1002/bit.28646.

The authors are requested to describe the contributions of their work compared to the mentioned references.

Response 4:

We appreciate the reviewers for pointing out these representative related studies. We acknowledge that the aforementioned literature has made important contributions to Raman spectroscopy-based modeling and deep learning methods. However, our work demonstrates uniqueness and innovations in the following aspects:

The cited studies mainly focus on biopharmaceutical cell culture monitoring, typically relying on large-scale process Raman data. In contrast, our research specifically addresses the quantitative analysis of the natural drug active component Tilianin, particularly establishing models under small-sample and powdered-sample conditions. This application scenario is relatively rare in existing literature and presents challenges including subtle spectral variations and limited data availability.

Compared to previous approaches using complex architectures like variational autoencoders (VAE) and semi-supervised CNNs, our proposed custom network structure RSAQN offers the advantages of lightweight architecture and stable performance, making it particularly suitable for drug detection tasks with limited data. Additionally, we emphasize model interpretability, which is especially crucial in practical pharmaceutical analysis.

We have added comparative analyses with these references in Chapter 4 to further clarify the marginal contributions of this study.

Comment 5:

In many places in the text, the word “And” is used right after the end of a finished sentence. Please correct/remove them throughout the manuscript.

Response 5:

We sincerely appreciate the reviewers' guidance on grammatical conventions. We have conducted a systematic review of the entire manuscript and corrected all improper uses of sentence-initial 'And'.

Comment 6:

In section 2.1 it is written “Weigh the desired mass of Tilianin powder on a balance and transfer it to a beaker. Using a pipette measure less than the target volume of methanol into the beaker. Heat gently and stir well using a glass rod until Tilianin is completely dissolved. Transfer the dissolved Tilianin solution to a volumetric flask. Use a standard volumetric

flask to finalize the volume and rinse the residual material several times to ensure that all Tilianin enters the solution system.” This type of writing is not a common practice in

academic writing and needs a major revision. Please revise these sentences throughout the text accordingly.

Response 6:

We sincerely appreciate the reviewers' valuable suggestions regarding the methodological presentation. We have comprehensively revised Section 2.1 and other experimental description sections, transforming instructional language into objective academic discourse.

Comment 7:

In section 2.2, the authors are needed to explain how the offline measurements are labeled with the Raman spectra in the training dataset.

Response 7:

We appreciate your comments. The mean values from triplicate measurements were used as spectral labels to accurately determine Tilianin concentrations, providing ground truth references for the Raman spectral data. The relative standard deviation (RSD < 0.8%) was calculated through three repeated measurements to ensure data precision. The corresponding modifications have been highlighted in red in the manuscript.

Comment 8: Section 2.3, de-meaned detrended

Response8 : We sincerely appreciate the reviewers' precise suggestions regarding our grammar. We have implemented the corresponding revisions in Section 2.3.

Comment 9:

In section 2.3 appropriate references are required for the baseline removal. Some of those references are:

- Eilers, P., & Boelens, H. F. M. (2005). Baseline correction with asymmetric least squares smoothing (Leiden University Medical Centre Report).

- Cai,Y., Yang, C., Xu, D., & Gui,W. (2018). Baseline correction for Raman spectra using penalized spline smoothing based on vector tra

---

## [Decision Letter · Decision Letter 1]

Study on the quantitative analysis of Tilianin based on Raman spectroscopy combined with deep learning

PONE-D-24-58474R1

Dear Dr. Pan,

We’re pleased to inform you that your manuscript has been judged scientifically suitable for publication and will be formally accepted for publication once it meets all outstanding technical requirements.

Kind regards,

Clara Sousa

Academic Editor

PLOS ONE

Reviewers' comments:

Reviewer's Responses to Questions

**Comments to the Author**

Reviewer #1: All comments have been addressed

2. Is the manuscript technically sound, and do the data support the conclusions?

Reviewer #1: Yes

3. Has the statistical analysis been performed appropriately and rigorously?

Reviewer #1: Yes

4. Have the authors made all data underlying the findings in their manuscript fully available?

Reviewer #1: Yes

5. Is the manuscript presented in an intelligible fashion and written in standard English?

Reviewer #1: Yes

Reviewer #1: I believe the authors have addressed the comments appropriately and this manuscript can be considered for publication.

**Do you want your identity to be public for this peer review?** For information about this choice, including consent withdrawal, please see our Privacy Policy

Reviewer #1: No

---

## [Editor Report · Acceptance letter]

PONE-D-24-58474R1

PLOS ONE

Dear Dr. Pan,

I'm pleased to inform you that your manuscript has been deemed suitable for publication in PLOS ONE. Congratulations! Your manuscript is now being handed over to our production team.

Kind regards,

on behalf of

Dr. Clara Sousa

Academic Editor

PLOS ONE